# Predictors of mental health during the Covid-19 pandemic in the US: Role of economic concerns, health worries and social distancing

**Fabrice Kämpfen**[1]*, **Iliana V. Kohler**[1], **Alberto Ciancio**[2], **Wändi Bruine de Bruin**[3,4,5,6], **Jürgen Maurer**[2], **Hans-Peter Kohler**[1]

**1** Population Studies Center and Department of Sociology, University of Pennsylvania, Pennsylvania, PA, United States of America, **2** Department of Economics, University of Lausanne, Lausanne, Switzerland, **3** Sol Price School of Public Policy, University of Southern California, Los Angeles, CA, United States of America, **4** Dornsife Department of Psychology, University of Southern California, Los Angeles, CA, United States of America, **5** Schaeffer Center for Health Policy and Economics, University of Southern California, Los Angeles, CA, United States of America, **6** Center for Economic and Social Research, University of Southern California, Los Angeles, CA, United States of America

* kampfenf@sas.upenn.edu

**Data Availability Statement:** Survey and data are publicly available (UAS, 2020). Understanding America Study - March 2020 Monthly Survey;

## Abstract

Despite the profound health and economic implications of Covid-19, there is only limited knowledge to date about the role of economic concerns, health worries and social distancing for mental health outcomes during the pandemic. We analyze online survey data from the nationally representative "Understanding America Study" (UAS) covering the period of March 10-31st 2020 (sample size: 6,585). Mental health is assessed by the validated PHQ-4 instrument for measuring symptoms of depression and anxiety. About 29% (CI:27.4-.30.4%) of the US adult population reported some depression/anxiety symptoms over the study period, with symptoms deteriorating over the month of March. Worsening mental health was most strongly associated with concerns about the economic consequences of the pandemic, while concerns about the potential implications of the virus for respondents' own health and social distancing also predicted increases in symptoms of depression and anxiety during the early stages of the pandemic in the US, albeit less strongly. Our findings point towards the possibility of a major mental health crisis unfolding simultaneously with the pandemic, with economic concerns being a key driving force of this crisis. These results highlight the likely importance of economic countermeasures and social policy for mitigating the impact of Covid-19 on adult mental health in the US over and above an effective public health response.

## Introduction

Among the myriad of consequences that the Covid-19 pandemic has on the health care, economic and social spheres in the US and worldwide, experts and policy makers are increasingly urging to consider the mental health consequences of the pandemic [1, 2]. Some observers

2020. Available from: https://uasdata.usc.edu/index.php.

**Funding:** We gratefully acknowledge the support of the Population Aging Research Center (PARC) and Population Studies Center (PSC) at the University of Pennsylvania, which are funded by the U.S. National Institutes of Health through grants NIA P30,AG12836 and NICHD R24 HD044964 respectively. The project described in this paper relies on data from survey(s) administered by the Understanding America Study (UAS), which is maintained by the Center for Economic and Social Research at the University of Southern California (USC). The content of this paper is solely the responsibility of the authors and does not necessarily represent the official views of USC or UAS. The collection of the UAS Covid-19 survey data is supported in part by the Bill & Melinda Gates Foundation, the National Institute on Aging (U01AG054580), and the National Science Foundation (#2028683). Wändi Bruine de Bruin was partially supported by the Swedish Riksbankens Jubileumsfond Program on Science and Proven Experience 'Science and Proven Experience'.

**Competing interests:** The authors have declared that no competing interests exist.

even went as far as calling mental illness resulting from Covid-19 the "inevitable" next pandemic [3]. The importance and urgency to address the short and long term aftermaths of Covid-19 on individual and population level mental health have been outlined in a recent policy brief issued by the United Nations [4]. Despite this compelling need to generate knowledge about the Covid-19 impact on mental health of individuals and populations, research on the mental health consequences of the pandemic, especially in the US, is still sparse and the determinants and sociodemographic patterns of mental well-being during Covid-19 are still not well documented. Prior evidence suggests that the experience of large scale disasters is associated with increases in depression and anxiety, post-traumatic stress disorder (PTD), and a broad range of other mental disorders [1, 5, 6]. In the context of Covid-19, these mental health implications may be amplified by factors such as increased uncertainty related to individual's own health because of the exposure to a new highly infectious disease [6–8], the profound economic consequences of the pandemic in the US and globally, the implementation of lockdown measures [9–12], resulting in the practice of prolonged social distancing [1, 13–16].

In this study, we investigate the consequences of the Covid-19 pandemic on mental health in the US between March 10th and March 31st, 2020, the early period of the pandemic when many US residents began to realize that Covid-19 was about to fundamentally affect their lives [9]. Using nationally representative population-based data for the US adult population from the Understanding America Study [17], we focus on three potential pathways through which the pandemic can affect mental health: uncertainties and perceptions related to the immediate impact of the coronavirus on own health, concerns about the economic consequences related to the pandemic and impact of practicing social distancing as a measure to contain the spread of the virus.

Uncertainties about the actual virus prevalence, its contagiousness and pathways of transmission, accompanied by difficulties to obtain testing and the lack of an effective treatment, can result in high perceived health threat and unpredictability of the real magnitude and impact of the disease. Although individuals may objectively face similar health risks, perceptions of these risks as well as information and knowledge about Covid-19 differ between individuals [18], potentially generating considerable differences in mental health responses. With all these uncertainties gravitating around the Covid-19 pandemic [8], risk perceptions and knowledge therefore can play a significant role in mental health outcomes and can drive depression and anxiety levels, to the point that the psychological distress about the disease can be more fearful than the disease itself [6, 7]

The large negative economic consequences of the various measures taken to contain the spread of Covid-19 are other potentially important drivers of mental health deterioration during the course of the pandemic in the US. Economic and financial security have long been recognized as important factors for mental well-being. [19–21] With the economy mostly shutting down and the unprecedented rise of unemployment, the immediate and long-term economic uncertainties are huge and individuals in the US are struggling to project themselves in the future and to secure an income for the coming months. [9] These economic uncertainties and challenges are likely exacerbated in the US by the weaker social safety net compared to other high-income countries. [22] With its major shock to the global economy, the Covid-19 pandemic is sending a massive number of individuals to unemployment and economic instability, making these individuals particularly vulnerable and prone to mental health disorders. [10–12]

Social distancing, which is a commonly implemented measure to reduce the spread of a virus, may also pose a mental health challenge for many people. Mass quarantine and shelter in place orders imposed throughout the US during the month of March to curtail the spread of Covid-19 have sent millions of individuals home in isolation. [16] While these steps may

contribute to flattening the curve of new infections, the lack of social interaction may have an impact on mental health due to feelings of loneliness and isolation. [1, 13–15]

In this analysis, we investigate the impact of the Covid-19 pandemic on mental health in the US between March 10th and March 31st, 2020, the early period of the pandemic when many US residents began to realize that Covid-19 was about to fundamentally affect their lives [9]. Using nationally representative population-based data for the US adult population from the Understanding America Study [17], we investigate how mental health is associated with uncertainties and perceptions related to the immediate impact of the coronavirus on own health, concerns about the economic consequences related to the pandemic and the practicing of social distancing.

## Methods

### Understanding America Study (UAS)

Our analysis utilizes the Covid-19 focused questionnaire of the UAS [17], implemented between March 10-31, 2020. UAS is a nationally representative probability-based Internet panel of approximately 8,500 respondents administered in English and Spanish. As part of its address-based study recruitment, UAS provides Internet access and a tablet to all panel members who may otherwise not be able to participate in the study [23]. In our study sample, 4.4% of the respondents were provided with an Internet-connected tablet at the time of recruitment in order to address the "digital divide" between different population groups (S1 Table). A comparison of UAS data with data from the Current Population Survey (CPS) and the Health and Retirement Study (HRS) shows that UAS lines up well with the CPS on a number of common variables, and matches the quality of the HRS, a traditional survey considered as the gold standard in social research [24, 25]. Additional details about UAS are provided in S1 Appendix. By March 31st 2020, out of the 8,815 participants to whom the questionnaire was fielded, 6,885 individuals (response rate: 78.1%) completed the survey. There were no significant differences in terms of age, gender and education between individuals who did and who did not complete the survey (S2 Table), even if non-completion was significantly higher among married UAS participants ($\chi^2$ = 5.659, p-value = 0.017). We restrict our analysis to respondents who completed the survey on the same day they started it and for which we have non-missing information about their mental health characteristics, leaving us with a study sample of 6,585 respondents. The online survey was approved by USC's Institutional Review Board. Survey and data are publicly available [17].

### Measurement of mental health in UAS

Mental health, i.e., presence of depressive and anxiety symptoms, is assessed by the Patient Health Questionnaire-4 (PhQ-4) which has been validated in the US [26]. The PhQ-4 is an ultra-brief four-item scale for detecting depression and anxiety, which represent the most common mental disorders during periods of disasters and disease outbreaks [27, 28] and are often co-occurring [29–33]. Composed by four distinct questions that are answered on a four-point Likert scale ranging from 0 ("Not at all") to 3 ("Nearly every day"), this scale has internal reliability and construct validity and is a reliable instrument for screening for both depressive and anxiety symptoms outside of clinical settings [26, 34] (see S2 Appendix for more details on PhQ-4). PhQ-4 scores ranged from 0 to 12, and we categorized respondents as having no depression/anxiety symptoms if their score was 0, 1 or 2, mild depression/anxiety symptoms if their score ranged from 3 to 5, moderate depression/anxiety symptoms if they scored 6 to 8 and severe depression/anxiety symptoms if their score was 9 or higher [26].

Importantly, the UAS provides information on the respondents' mental health prior to the Covid-19 outbreak, albeit based on a different survey item. Between 2018 and 2020, panel participants were asked whether they agree or not to the statement: "I am someone who is depressed, blue", with five permissible answers ranging from "strongly disagree" to "agree strongly". Derived from the Big Five Personality Scale [35] and not fully comparable with the PhQ-4 scale, we used this information as a measure for the presence of depressive symptoms to control for any underlying differences in mental health characteristics among the UAS respondents prior to the outbreak of Covid-19 in the US.

## Measurement of Covid-19 related risks, behaviors and events

In addition to sociodemographic characteristics such as sex, age, education, race and marital status at the time of the interview (see descriptive statistics in S1 Table), UAS elicited respondents' perceptions that specific Covid-19-related events will occur, using a validated visual linear scale ranging from 0 to 100 [36]. The survey asked respondents to rate their probability of getting infected with coronavirus in the next three months and the probability of dying in case of infection. We generated a measure of respondents' perceived risk of dying from Covid-19 by multiplying these two probabilities. To measure economic concerns about the Covid-19 impact, survey participants were asked to rate the chances that they will run out of money within the next three months. The respective questions can be found in S2 Appendix. UAS also asked participants whether they have reduced their social life activities because of the pandemic. To measure social distancing, we generated a dichotomous variable that takes the value of one if respondents stated that they have canceled or postponed travel for pleasure, and/or canceled or postponed personal or social activities, and/or avoided public space gatherings or crowds and/or eating at a restaurant (Cronbach's $\alpha = 0.804$, indicating good reliability of the items to measure the same construct [37]).

The period of our study, March 10–31, 2020, is characterized by rapid increase of reported Covid-19 infections and related deaths throughout the US on a daily basis, with the North-East being most affected by the pandemic at that time. To assess the impact of the increasing caseloads in Covid-19 infections and deaths on mental health in the US, we matched to our UAS sample publicly available daily data on the total number of cases and deaths in the US during our study period in March. These Covid-19 data is provided by state and local governments, and health departments and the information is collected and compiled by New York Times on a daily basis. [38] The match of the data was based on the day when the UAS respondents completed the survey.

## Statistical analysis

To explore the pathways through which the Covid-19 pandemic affects mental health in the US, we estimated separate ordered probit regression models with three different explanatory variables that reflect: 1) individual's economic concerns induced by the pandemic (measured by the perceived probability of running out of money on a continuous scale); 2) distress due to the immediate health impacts of the virus (measured by the probability of dying from Covid-19 on a continuous scale); and 3) the influence of practiced social distancing. The outcome variable in all three models is a categorical variable that indicates the presence and severity of depression/anxiety symptoms measured by the PhQ-4, as defined above, where higher values indicate presence of more severe symptoms.

Usual standardization to compare the strength of these associations with mental health is inadequate because social distancing is measured by a dichotomous variable. Hence, to ensure a more meaningful comparison between the three explanatory variables, we subtracted the

mean and re-scaled the two continuous variables by dividing them by $\gamma = 2.103$ times their respective standard deviation so that the coefficients of the continuous variables correspond in magnitude to a marginal increase in the dichotomous variable from 0 to 1 (see S3 Appendix for more details).

Our econometric specifications control for sex, educational level (binary variables for whether the respondents completed high school, had some college education or completed college or more), race and whether the respondent was married at the time of the interview. Prior research shows that Covid-19 risk perceptions and mental health differ by age [39], we therefore control for age by including its second order polynomial in our specifications to account for any possible non-linear relationships between age and our dependent variables. Our models also controlled for state fixed effects, the self-reported depression level characteristics prior to the pandemic and the year and month when this measure was collected. The model specifications also included time fixed effects (in days) to capture the influence of aggregate effects, such as public announcements at the federal level or national news, that are shared among UAS respondents on a given day. Our analysis used post-stratification weights, generated through a raking algorithm, to align the sample to the US adult population in terms of gender, race/ethnicity, age, education and geographic location (see https://uasdata.usc.edu/page/Weights and [24]). Analysis was performed with StataSE 14.

## Results

Table 1 summarizes the weighted distribution characteristics of the main variables used in our analysis. The weighted mean of the PhQ-4 score in the US was about 1.9, with 71.1% of respondents reporting no depressive/anxiety symptoms, while 17.8% reported mild depressive/anxiety symptoms; 6.6% of the sample had moderate and 4.6% experienced severe depressive/anxiety symptoms.

During the period March 10-31st, 2020, respondents reported a 15.8% probability to run out of money within three months because of the pandemic's impact. This is higher than the population mean reported previously for the period March 10-16th (13.2%) using the UAS survey [9], indicating a deterioration in the perceived economic perspectives in the US during the month of March. Similarly, our sample showed a weighted mean of perceived risk of dying from Covid-19 of about 4.3%, which is also higher than the previously reported mean (3.8%) based on the UAS [9]. Table 1 also shows that less than two thirds of the US population were practicing social distancing during the study period.

Fig 1 shows weighted means of PhQ-4 score on the left and weighted proportions of the US population that has a least some depressive/anxiety symptoms on the right over the period March 10-31. By applying post-stratification weights, the weighted means and proportions are representative of the US population for each particular time period on the x-axis. Reports of depression/anxiety symptoms increased over time in March, reaching the highest point in the last week of the month. This increase was not driven by persons with higher depression level completing the online survey later in the month, as the trend is not increasing when we consider depression level characteristics prior to the Covid-19 pandemic instead of the PhQ-4 mental health measure collected in March (see S1 Fig).

We also investigated the associations between PhQ-4 and the rapidly increasing number of Covid-19 infection cases and deaths during the period March 10-31, 2020 (Table 2). A one-unit increase in the log number of Covid-19 cases resulted on average in about 0.090 increase in the latent mental health ordered probit index (z-score). This increase corresponded to a drop of 2.6 percentage points in the probability of reporting no depressive/anxiety symptoms, whereas the probability of reporting mild, moderate and sever symptoms increases by 1.2, 0.7

**Table 1. Summary statistics (*N* = 6, 585).**

| Variables | Scale | Pop. mean March 10-31 | Pop. std deviation March 10-31 | Pop. mean March 10-16 | Pop. std deviation March 10-16 |
|---|---|---|---|---|---|
| *Outcome variables* | | | | | |
| PhQ-4 score | From 0 to 12 | 1.944 | 2.829 | 1.910 | 2.861 |
| No depression/anxiety symptoms | Yes (1)/no (0), PhQ-4 score 0-2 | 0.711 | 0.453 | 0.723 | 0.448 |
| Mild depression/anxiety symptoms | Yes (1)/no (0), PhQ-4 score 3-5 | 0.178 | 0.382 | 0.165 | 0.371 |
| Moderate depression/anxiety symptoms | Yes (1)/no (0), PhQ-4 score 6-8 | 0.066 | 0.247 | 0.064 | 0.244 |
| Severe depression/anxiety symptoms | Yes (1)/no (0), PhQ-4 score 9+ | 0.046 | 0.209 | 0.049 | 0.215 |
| *Explanatory variables of interest* | | | | | |
| Perceived risk of running out of money | Continuous scale from 0 to 1 | 0.158 | 0.261 | 0.132 | 0.237 |
| Perceived risk of dying from Covid-19 | Continuous scale from 0 to 1 | 0.043 | 0.093 | 0.038 | 0.090 |
| Self-reported social distancing | Yes (1)/no (0) | 0.655 | 0.475 | 0.577 | 0.494 |
| *Depression level characteristics prior to the pandemic* | | | | | |
| I am someone who is depressed, blue | Strongly disagree (1)–Agree strongly (5) | 2.210 | 1.285 | | |
| Strongly disagree | Yes (1)/no (0) | 0.438 | 0.496 | | |
| Disagree a little | Yes (1)/no (0) | 0.171 | 0.377 | | |
| Neither Agree nor Disagree | Yes (1)/no (0) | 0.179 | 0.383 | | |
| Agree a little | Yes (1)/no (0) | 0.165 | 0.372 | | |
| Agree strongly | Yes (1)/no (0) | 0.046 | 0.210 | | |

*Note*: Source of Data: "Understand America Study" (UAS), survey 230, collected between March 10 and March 31, 2020. All statistics use sample weights to make the survey representative of the U.S. population aged 18 and older. Depression level characteristics prior to the pandemic are based on the Big Five Personality Scale and are derived from the question: "I see myself as someone who is depressed, blue". This question was asked in UAS survey 121, fielded between Jan 2018 and March 2020. We consider someone as having practised social distancing if a respondent "canceled or postponed travel for pleasure", "canceled or postponed personal or social activities", "avoided public spaces, gatherings or crowds" or "avoided eating at restaurant". "Pop." is short for population and "std." is short for standard.

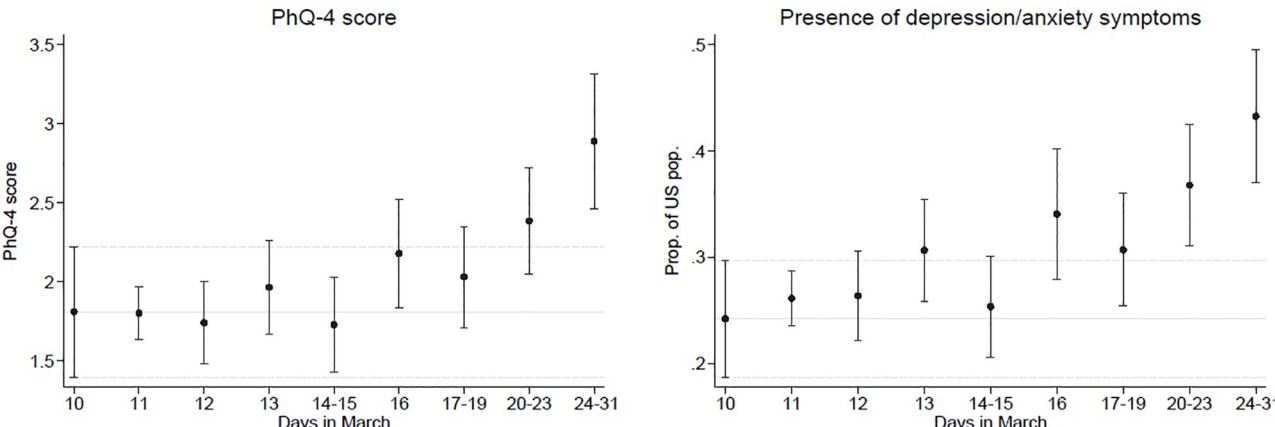

**Fig 1. Changes in the mental health (PhQ-4) from March 10th to March 31st.** *Notes*: Source of Data: "Understand America Study" (UAS), survey 230, collected between March 10 and March 31, 2020. Plots shows weighted means, along with 95% confidence intervals, of PhQ-4 score on the left and weighted proportions of the US population that has at least some depressive/anxiety symptoms on the right. We use post-stratification weights so that the weighted means and proportions are representative of the US population for *each particular time period* on the x-axis.

**Table 2. Associations with the raising number of Covid-19 cases and deaths in the US on mental health score (PhQ-4).**

| | PhQ-4 | PhQ-4 Female | PhQ-4 Male | PhQ-4 No college | PhQ-4 College | PhQ-4 White | PhQ-4 Non-white |
|---|---|---|---|---|---|---|---|
| | (1) | (2) | (3) | (4) | (5) | (6) | (7) |
| **Number of cases in US (log)** | 0.090*** | 0.074*** | 0.113*** | 0.070** | 0.137*** | 0.093*** | 0.082 |
| | [0.044,0.136] | [0.027,0.122] | [0.039,0.187] | [0.014,0.126] | [0.080,0.195] | [0.046,0.141] | [-0.017,0.182] |
| Observations | 6514 | 3790 | 2724 | 3844 | 2670 | 5084 | 1430 |
| **Number of deaths in US (log)** | 0.104*** | 0.086*** | 0.131*** | 0.080** | 0.157*** | 0.107*** | 0.099* |
| | [0.051,0.157] | [0.031,0.141] | [0.044,0.218] | [0.015,0.146] | [0.090,0.224] | [0.050,0.163] | [-0.018,0.215] |
| Observations | 6514 | 3790 | 2724 | 3844 | 2670 | 5084 | 1430 |

*Notes*: Results of weighted ordered probit regressions with cluster robust confidence intervals at the state level in brackets,

* $p < 0.1$,

** $p < 0.05$,

*** $p < 0.01$.

The number of cases and deaths in the US is transformed using log. Specifications include state fixed effects. The list of control variables includes: sex, age and age$^2$, educational level (binary variable for each category), race and whether the respondent was married at the time of the interview. We control for depression level characteristics prior to the Covid-19 pandemic, along with the year and month when this measure was collected. We use sample weights to make the survey representative of the U.S. population aged 18 and older. Data source: "Understanding America Study" (UAS) collected between March 10 and March 31, 2020.

and 0.7, respectively (S3 Table). This association was particularly strong for males and college graduates. Similar patterns were also estimated for the impact of the number of deaths in the US, where a one-unit increase in the log number of Covid-19 deaths led on average to an increase of about 0.104 in the latent mental health index, indicating higher levels of depression/anxiety, with this effect being again stronger for males and college graduates (Table 2 columns 3 and 5).

High expected probability of running out of money is positively associated with the PhQ-4 mental health score (Table 3, Column 1). Similarly, perceived risks of dying from Covid-19

**Table 3. Associations with mental health score (PhQ-4).**

| | PhQ-4 | PhQ-4 | PhQ-4 | PhQ-4 | PhQ-4 rescaled |
|---|---|---|---|---|---|
| | (1) | (2) | (3) | (4) | (5) |
| Perceived risk of running out of money | 0.850*** | | | 0.724*** | **0.398 ***** |
| | [0.672,1.028] | | | [0.537,0.911] | **[0.295,0.500]** |
| Perceived risk of dying from Covid-19 | | 1.336*** | | 0.846*** | **0.165 ***** |
| | | [0.907,1.766] | | [0.405,1.286] | **[0.079,0.251]** |
| Self-reported social distancing | | | 0.307*** | 0.241*** | **0.241 ***** |
| | | | [0.204,0.410] | [0.136,0.347] | **[0.136,0.347]** |
| Observations | 6436 | 6377 | 6434 | 6368 | **6368** |

*Notes* Results of weighted ordered probit regressions with robust confidence intervals in brackets,

* $p < 0.1$,

** $p < 0.05$,

*** $p < 0.01$.

Specifications include state and time fixed effects. The list of control variables includes: sex, age and age$^2$, educational level (binary variable for each category), race and whether the respondent was married at the time of the interview. We control for depression level characteristics prior to the Covid-19 pandemic, along with the year and month when this measure was collected. We use sample weights to make the survey representative of the U.S. population aged 18 and older. "Perceived risk of running out money" and "Perceived risk of dying from Covid-19" in Column 5 have been standardized by subtracting their respective mean and dividing by $\gamma \times$ their standard deviations. Data source: "Understanding America Study" (UAS) collected between March 10 and March 31, 2020.

**Table 4. Associations with mental health score (PhQ-4).**

| | PhQ-4 Female | PhQ-4 Male | Difference Male vs Female | PhQ-4 No college | PhQ-4 College | Difference No college vs College | PhQ-4 White | PhQ-4 Non-white | Difference White vs Non-white |
|---|---|---|---|---|---|---|---|---|---|
| | (1) | (2) | p-value | (3) | (4) | p-value | (5) | (6) | p-value |
| Perceived risk of running out of money (std) | 0.311*** | 0.541*** | 0.021 | 0.416*** | 0.371*** | 0.705 | 0.367*** | 0.466*** | 0.410 |
| | [0.186,0.435] | [0.380,0.701] | | [0.296,0.535] | [0.169,0.573] | | [0.256,0.478] | [0.252,0.680] | |
| Perceived risk of dying from Covid-19 (std) | 0.222*** | 0.073 | 0.090 | 0.165*** | 0.164** | 0.988 | 0.217*** | -0.007 | 0.040 |
| | [0.120,0.325] | [-0.070,0.216] | | [0.064,0.266] | [0.018,0.309] | | [0.121,0.312] | [-0.199,0.186] | |
| Self-reported social distancing | 0.212*** | 0.267*** | 0.600 | 0.202*** | 0.338*** | 0.190 | 0.237*** | 0.264** | 0.854 |
| | [0.078,0.346] | [0.105,0.429] | | [0.070,0.334] | [0.178,0.498] | | [0.125,0.350] | [-0.000,0.528] | |

*Notes*: Results of weighted ordered probit regressions with robust confidence intervals in brackets,

* $p < 0.1$,

** $p < 0.05$,

*** $p < 0.01$.

Specifications include state and time fixed effects. The list of control variables includes: sex, age and age$^2$, educational level (binary variable for each category), race and whether the respondent was married at the time of the interview. We control for depression level characteristics prior to the Covid-19 pandemic, along with the year and month when this measure was collected. We use sample weights to make the survey representative of the U.S. population aged 18 and older. "Perceived risk of running out money" and "Perceived risk of dying from Covid-19" have been standardized by subtracting their respective mean and dividing by $\gamma \times$ their standard deviations. Data source: "Understanding America Study" (UAS) collected between March 10 and March 31, 2020.

and social distancing (Columns 2 and 3) are also strongly correlated with higher levels of depression/anxiety. Column 4 shows that when all of the three explanatory variables are included in the same model specification, they remain strongly associated with the PhQ-4 mental health score, "independently" from each other. However, because of the different scales in which these variables are measured, the magnitude of their associations cannot be directly compared. Column 5 allows a direct comparison of the coefficients after standardizing the continuous variables as explained above. The association of the probability of running out of money with mental well-being is stronger than the two other variables capturing alternative pathways affecting mental health during the pandemic. Social distancing shows the second-strongest association, while perceived risks of dying from the virus has the weakest association with mental health among the three. The result of a one-sided z-test rejects the null hypothesis that the coefficient associated with social distancing is equal or larger than the coefficient associated with running out of money (p-value = 0.027). Average marginal effects reported in S4 Table show that a $\gamma$ standard deviation increase in the perceived probability of running out of money because of Covid-19 led to a 11.1 percentage points decrease in the probability of reporting no depressive/anxiety symptoms, whereas it increases the probability of showing mild, moderate and severe symptoms by 5.1, 3.0 and 3.1, respectively. The magnitude of the average marginal effects of the two other explanatory variables are about half of those of the economic uncertainties, with social distancing having larger associations with mental health than perceived health concerns.

Table 4 reveals important heterogeneity in these patterns, with perceived economic uncertainty being particularly important for males (Columns 1 and 2), whereas concerns about own health being less important for males and non-white individuals (Columns 1,2, 5 and 6). The association of social distancing with PhQ-4 mental health score however appears to be relatively similar across the various sociodemographic groups. Results including the interaction

terms between our three main independent variables and the various sociodemographic
groups are presented in S5 Table.

## Discussion

We assessed factors associated with mental health of adults age 18+ years during the early out-
break of the Covid-19 pandemic, March 10-31, 2020 in the US. During this period, the virus
spread rapidly in the US, with confirmed infection cases and deaths rising exponentially, from
1018 to 187,962 and from 31 to 3630, respectively [38]. We used the clinically validated PhQ-4
instrument to assess the presence of depressive and anxiety symptoms in a nationally represen-
tative sample of US adults aged 18+ years. To our best knowledge, our study is one of the first
to document the decline in US adults' mental health during the early period of the Covid-19
pandemic.

In March, symptoms of depression and anxiety among US adults were increasing and this
worsening of mental health was associated with the increasing number of confirmed Covid-19
cases and related deaths in the US. However, increased caseloads was not the primary driving
force behind this deterioration of mental well-being among US adults. Rather, concerns about
the economic consequences of the pandemic in the near future (i.e., three months from the
date of the interview) emerged as the strongest predictor of declining mental health. Worries
about the potential impacts of Covid-19 on own health (i.e., risk of infection and as result
increased risk of death), as well as the practice of social distancing played relatively smaller
roles in predicting poor mental well-being during the early stages of the pandemic. We also
documented significant gender differences in the associations of mental health with perceived
economic and health risks: Economic concerns are more strongly associated with worse men-
tal health in men than women, whereas the association of mental health with concerns related
to own health outcomes is stronger in women than men, which may partly reflect differences
in gender roles and behaviors [40–42]. Moreover, we only find a positive association between
perceived risk of dying from Covid-19 and mental health decline for white respondents, with
no such association among non-whites. These ethnic/racial differences may be related to cor-
responding differences in the socioeconomic impact of Covid-19 across racial/ethnic groups
with economic concern/worries being a main driver of depression among minorities [43].

### Limitations

The evidence of mental health issues arising during the Covid-19 pandemic emphasizes the
importance of future research on this topic and specifically the need to investigate the long-
term effects of the pandemic on mental health outcomes. For instance, prolonged social dis-
tancing measures may result in stronger impact on mental well-being at later stages of the pan-
demic compared to the weak relationship established in this study. In addition, the pandemic
may also have introduced an economic crisis, which will negatively affect mental health for
those affected. [44] On the other hand, prior evidence points to the possibility that people may
adapt to crisis situations over time, suggesting mental health might improve in the long term
[45]. In addition, with Covid-19 tests becoming increasingly available, the positive outlooks
for the developments of potential treatments and vaccines, and positive news coming from
countries that were able to contain the spread of the pandemic might however attenuate the
health concerns associated with mental health. The long-term consequences of the Covid-19
pandemic on mental health are therefore of particular research interest especially given the
dramatic increases in unemployment in the US that have occurred after the data collection for
this study was completed. Given the relatively weak social safety net in the US compared to
European high-income countries, the importance and urgency to address the aftermaths of

Covid-19 on individual and population level mental health is and will be all the more critical to address in the US.

Our correlational data did not warrant causal conclusions. Although it is possible that economic concerns, health worries and social distancing resulted in declining mental health, it is also possible that having worse mental health led to more rumination about economic concerns and health worries, as well as engaging more in the practice of social distancing. While our sample is representative for the US adult population, our estimates did not include minors below age 18, for whom economic concerns may not be at the forefront, but whose mental well-being maybe as well affected by the disruption of their daily routines, schooling, extra-curriculum activities, exposure to stress in the household and increased domestic abuse and violence. PhQ-4 may capture only probable depression/anxiety or psychological distress in response to the abnormality of the Covid-19 pandemic and its implications as opposed to clinical diagnostic, which does not reduce the importance of our findings for public health policy. Although most adults report only mild to moderate levels of depression and anxiety, these may nonetheless affect many important outcome such as work productivity and savings [46, 47].

## Conclusion

Our evidence on declining mental health during the early period of the Covid-19 pandemic highlights the importance of imminent mental health challenges of the pandemic among US adults, which may further increase as Covid-19 continues to unfold. As a result, there appears to be an increasing need for prevention and mental health services as a consequence of the pandemic, requiring a parallel strengthening of such efforts during the pandemic. Policy makers, health care providers and social workers should therefore plan for a substantial increase in service needs since the ramifications of the Covid-19 pandemic will be likely felt for an extended period of time beyond getting the virus under control and the curve flattened. In addition, our findings highlight the major importance of economic considerations for US adults' mental health early in the pandemic over and above the evident health concerns and challenges associated with social distancing. Our results suggest a considerable role for economic countermeasures and social policy for mitigating the economic impacts of the Covid-19 pandemic on US adults' livelihoods and, thereby, helping to protect their mental health and well-being through this unfolding pandemic.

## Supporting information

**S1 Fig. Associations with depression level characteristics prior to the Covid-19 pandemic.** *Notes*: Source of Data: "Understand America Study" (UAS), surveys 121 and 230. UAS survey 121 was fielded between January 2018 and March 2020 and survey 230 was collected between March 10 and March 31, 2020. Plot on the left shows weighted proportions of the US population, along with 95% confidence intervals, that strongly disagree to the statement: "I see myself as someone who is depressed, blue". Plot on the right shows weighted proportions of the US population that strongly disagree or disagree to the same statement. We use post-stratification weights so that the weighted means are representative of the US population for *each* particular time period on the *x–axis*.
(TIF)

**S1 Appendix. Understanding America Study (UAS).**
(PDF)

**S2 Appendix. The PhQ-4 and Covid-19-related questions.**
(PDF)

**S3 Appendix. γ-standardization of the continuous variables.**
(PDF)

**S1 Table. Descriptive statistics—additional variables.**
(PDF)

**S2 Table. Differences between UAS responders and non-responders.**
(PDF)

**S3 Table. Associations with Covid-19 cases and related number of deaths and mental health score (PhQ-4)—average marginal effects.**
(PDF)

**S4 Table. Associations with mental health score (PhQ-4)—average marginal effects.**
(PDF)

**S5 Table. Associations with mental health score (PhQ-4).**
(PDF)

## Author Contributions

**Conceptualization:** Fabrice Kämpfen, Iliana V. Kohler, Alberto Ciancio, Hans-Peter Kohler.

**Data curation:** Fabrice Kämpfen, Alberto Ciancio.

**Formal analysis:** Fabrice Kämpfen, Alberto Ciancio.

**Methodology:** Fabrice Kämpfen, Iliana V. Kohler, Alberto Ciancio, Hans-Peter Kohler.

**Writing – original draft:** Fabrice Kämpfen, Iliana V. Kohler, Alberto Ciancio.

**Writing – review & editing:** Fabrice Kämpfen, Iliana V. Kohler, Alberto Ciancio, Wändi Bruine de Bruin, Jürgen Maurer, Hans-Peter Kohler.

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
