## [Decision Letter · Decision Letter 0]

8 Sep 2020

PONE-D-20-20513

Predictors of mental health during the Covid-19 pandemic in the US: role of economic concerns, health worries and social distancing

PLOS ONE

Dear Dr. Kampfen,

Thank you for submitting your manuscript to PLOS ONE. After careful consideration, we feel that it has merit but does not fully meet PLOS ONE’s publication criteria as it currently stands. Therefore, we invite you to submit a revised version of the manuscript that addresses the points raised during the review process.

We look forward to receiving your revised manuscript.

Kind regards,

Young Dae Kwon, M.D., Ph.D.

Academic Editor

PLOS ONE

Journal Requirements:

2. In your Methods section, please provide additional information about the methodology. In particular, please specify the response rate, and how many participants are included in the present analysis; and whether you applied any additional exclusion criteria.

Reviewers' comments:

Reviewer's Responses to Questions

**Comments to the Author**

1. Is the manuscript technically sound, and do the data support the conclusions?

Reviewer #1: Yes

Reviewer #2: Yes

2. Has the statistical analysis been performed appropriately and rigorously? 

Reviewer #1: Yes

Reviewer #2: Yes

3. Have the authors made all data underlying the findings in their manuscript fully available?

Reviewer #1: Yes

Reviewer #2: Yes

4. Is the manuscript presented in an intelligible fashion and written in standard English?

Reviewer #1: Yes

Reviewer #2: Yes

5. Review Comments to the Author

Reviewer #1: - I think this article timely in the context of Covid-19 pandemic.

- The article demonstrated strongly association between concerns about the economic consequences of the pandemic and worsening mental health. If we assumed a reverse causality which means that the person who have worsening mental health might be likely to be a lower income person, the association among them might be overestimated.

- The Authors showed association between social distance and worsening mental health. If we assumed a reverse causality which means that worsening mental health might increase social distance, the association among them might be overestimated.

- I suggest the authors to describe the reverse causality from outcome variable to the control variables in detail.

Reviewer #2: -The subject of the thesis is timely.

-The data are comprehensive enough to capture the current situation of Covid-19.

-The analysis method is sophisticated.

-However, the results are too general.

In other words, it seems that the intellectual and logical understanding of the context of the variables are insufficient. In particular, there is not enough logical and theoretical correlation between visible economic anxiety in unprecedented uncertain situations, the threat of infectious diseases where information asymmetry exists invisibly, and social distance that inhibits everyday life and action. Therefore, it is expected that further studies will be supplemented over a longer period of time between variables.

6. PLOS authors have the option to publish the peer review history of their article (what does this mean?). If published, this will include your full peer review and any attached files.

Reviewer #1: No

Reviewer #2: No

---

## [Author Response · Author response to Decision Letter 0]

19 Oct 2020

Response letter to the reviewers’ comments on manuscript submission PONE-D-20-20513 Entitled “Predictors of mental health during the Covid-19 pandemic in the US: role of economic concerns, health worries and social distancing”

Dear Editor,

dear Reviewers:

Thank you very much for your valuable feedback on our manuscript PONE-D-20-20513. We appreciate the opportunity to revise and resubmit our manuscript to PLOS ONE. The provided comments raised some good points and were helpful to improve the manuscript. We hope that the revised paper will be acceptable for publication in PLOS ONE.

This letter outlines our changes in the manuscript in response to the reviewers’ comments and suggestions and provides specific answers to all issues raised in their reviews. For convenience, we first reproduce the reviewers’ comments and then provide corresponding answers after each comment in italics.

Reviewers' comments:

Reviewer #1: 

1) The article demonstrated strongly association between concerns about the economic consequences of the pandemic and worsening mental health. If we assumed a reverse causality which means that the person who have worsening mental health might be likely to be a lower income person, the association among them might be overestimated.

2) The Authors showed association between social distance and worsening mental health. If we assumed a reverse causality which means that worsening mental health might increase social distance, the association among them might be overestimated.

3) I suggest the authors to describe the reverse causality from outcome variable to the control variables in detail.

Authors’ response: Thank you for these three comments that refer to the issue of causality in our analyses. It is true that our analysis does not allow to uncover the causal effects of economic concerns, health worries and social distancing on mental health. As you correctly point out, there is likely to be a “reverse causality” at play in the sense that mental health condition might affect Covid-related perceptions and behaviors, even after controlling for past mental health characteristics as we do in our analysis. In other words, our control variables of interest are endogenous and no causal claims can be made. 

To address your comment, we have expanded our limitation section by adding a sentence that discusses the reverse causality issues you are referring to. We added the following statement:

“Our correlational data did not warrant causal conclusions. Although it is possible that economic concerns, health worries and social distancing resulted in declining mental health, it is also possible that having worse mental health led to more rumination about economic concerns and health worries, as well as engaging more in the practice of social distancing.” (p. 17/30)

We have also removed the causal language in our manuscript, such as “influence” and “implications”, and replaced these words with “associations”. 

Reviewer #2: 

1) The results are too general. In other words, it seems that the intellectual and logical understanding of the context of the variables are insufficient. In particular, there is not enough logical and theoretical correlation between visible economic anxiety in unprecedented uncertain situations, the threat of infectious diseases where information asymmetry exists invisibly, and social distance that inhibits everyday life and action. Therefore, it is expected that further studies will be supplemented over a longer period of time between variables.

Authors’ response: Thank you for this comment, which we have addressed in two ways. 

First, we have expanded our introduction section by providing more details on the reasons why one can expect the Covid-19 pandemic to have an effect on individual’s mental health:

“Uncertainties about the actual virus prevalence, its contagiousness and pathways of transmission, ac-companied by difficulties to obtain testing and the lack of an effective treatment, can result in high per-ceived health threat and unpredictability of the real magnitude and impact of the disease. Although in-dividuals may objectively face similar health risks, perceptions of these risks as well as information and knowledge about Covid-19 differ between individuals [18], potentially generating considerable differ-ences in mental health responses. With all these uncertainties gravitating around the Covid-19 pandem-ic [8], risk perceptions and knowledge therefore can play a significant role in mental health outcomes and can drive depression and anxiety levels, to the point that the psychological distress about the dis-ease can be more fearful than the disease itself.[6,7]

The large negative economic consequences of the various measures taken to contain the spread of Covid-19 are other potentially important drivers of mental health deterioration during the course of the pandemic in the US. Economic and financial security have long been recognized as important factors for mental well-being.[19-21] With the economy mostly shutting down and the unprecedented rise of unem-ployment, the immediate and long-term economic uncertainties are huge and individuals in the US are struggling to project themselves in the future and to secure an income for the coming months.[9] These economic uncertainties and challenges are likely exacerbated in the US by the weaker social safety net compared to other high-income countries.[22] With its major shock to the global economy, the Covid-19 pandemic is sending a massive number of individuals to unemployment and economic instability, making these individuals particularly vulnerable and prone to mental health disorders.[10-12]

Social distancing, which is a commonly implemented to reduce the spread of a virus, may also pose a mental health challenge. Mass quarantine and shelter in place orders imposed throughout the US during the month of March have sent millions of individuals home in isolation.[16] While these steps may contribute to flattening the curve of new infections, the lack of social interaction may reduce mental health due to feeling of loneliness and isolation.[1,13-15]” (page 3/30)

Secondly, we modified the first paragraph of the limitation section to stress the need for more research on the long-term impacts of Covid-19 on mental health. The paragraph we modified now reads as follows:

“The evidence of mental health issues arising during the Covid-19 pandemic emphasizes the importance of future research on this topic and specifically the need to investigate the long-term effects of the pandemic on mental health outcomes. For instance, prolonged social distancing measures may result in stronger impact on mental well-being at later stages of the pandemic compared to the weak relationship established in this study. In addition, the pandemic may also have introduced an economic crisis, which will negatively affect mental health for those affected [44]. On the other hand, prior evidence points to the possibility that people may adapt to crisis situations over time, suggesting mental health might improve in the long term [45]. In addition, with Covid-19 tests becoming increasingly available, the positive outlooks for the developments of potential treatments and vaccines, and positive news coming from countries that were able to contain the spread of the pandemic might however attenuate the health concerns associated with mental health. The long-term consequences of the Covid-19 pandemic on mental health are therefore of particular research interest especially given the dramatic increases in unemployment in the US that have occurred after the data collection for this study was completed. Given the relatively weak social safety net in the US compared to European high-income countries, the importance and urgency to address the aftermaths of Covid-19 on individual and population level mental health is and will be all the more critical to address in the US.” (page 17/30)

Authors’ response: N/A.

2. In your Methods section, please provide additional information about the methodology. In particular, please specify the response rate, and how many participants are included in the present analysis; and whether you applied any additional exclusion criteria.

Authors’ response: Our method section already included information about the response rate (page 3 line 42 contains the required information: “By March 31st 2020, out of the 8,815 participants to whom the questionnaire was fielded, 6,885 individuals (78.1%) completed the survey.)”. We also added the following sentence in the method section that details the exclusion restriction to our sample we applied. 

“We restrict our analysis to respondents who completed the survey on the same day they started it and for which we have non-missing information about their mental health characteristics, leaving us with a study sample of 6,585 respondents.”

We also added on the top of the summary statistics table the number of observations in our study sample. 

Some additional changes that we made:

We made some additional changes in the labels of the variables in the tables to be consistent with the labels and changes in the main text. These changes are clearly marked.

---

## [Editor Report · Decision Letter 1]

23 Oct 2020

Predictors of mental health during the Covid-19 pandemic in the US: role of economic concerns, health worries and social distancing

PONE-D-20-20513R1

Dear Dr. Kampfen,

We’re pleased to inform you that your manuscript has been judged scientifically suitable for publication and will be formally accepted for publication once it meets all outstanding technical requirements.

Kind regards,

Young Dae Kwon, M.D., Ph.D.

Academic Editor

PLOS ONE
---

## [Editor Report · Acceptance letter]

30 Oct 2020

PONE-D-20-20513R1 

Predictors of mental health during the Covid-19 pandemic in the US: role of economic concerns, health worries and social distancing 

Dear Dr. Kämpfen:

I'm pleased to inform you that your manuscript has been deemed suitable for publication in PLOS ONE. Congratulations! Your manuscript is now with our production department. 

Kind regards, 

on behalf of

Dr. Young Dae Kwon 

Academic Editor

PLOS ONE